# Oral health profiles in the population of older adults in Ecuador: An analysis of latent classes

Adriana Canessa-Rojas[1], Marite Falquez-Flor[2], Stephanie Gallegos-Caamñano[2], Marco Faytong-Haro [1,3]*

1 Facultad de Ciencias de la Salud and School of International Studies, Universidad Espíritu Santo, Samborondón, Guayas, Ecuador, 2 Ecuadorian Development Research Lab, Daule, Guayas, Ecuador, 3 Department of Sociology and Criminology and the Population Research Institute, Pennsylvania State University, University Park, Pennsylvania, United States

* mfaytong@uees.edu.ec

## Abstract

### Background

Ecuador's aging population highlights the importance of addressing oral health issues among older adults because of the potential negative impact on their quality of life, nutrition, and overall well-being. Understanding the oral health profiles and specific needs of this population is crucial for the development of effective prevention and dental care strategies. This study aimed to identify and characterize the oral health profiles of older adults in Ecuador.

### Methods

Data from the SABE 2009 survey were analyzed, encompassing responses from 5235 older adults from Ecuador. Oral health profiles were identified using oral health-related variables such as level of edentulism, the use of dentures, frequency of dental problems such as the change of food, among others. Additionally, a logistic regression analysis was conducted, in which sociodemographic and behavioral factors were assessed as predictors of the oral health profile membership of individuals.

### Results

The analysis revealed 8 distinct oral health profiles among older Ecuadorian adults, varying in oral health status, prevalence of dental problems, and oral hygiene levels. Among these profiles, the three latent classes with the highest marginal probabilities were: (1) individuals with no original teeth, possessing complete dentures, minimal dental problems, and high satisfaction (0.22); (2) individuals with many missing teeth, dentures, minimal problems, and high satisfaction (0.19); and (3) individuals with all teeth missing, wearing dentures, with significant problems and high satisfaction

**Data availability statement:** The dataset file analysed during the current study is available from https://anda.inec.gob.ec/anda/index.php/catalog/292.

**Funding:** The author(s) received no specific funding for this work.

**Competing interests:** The authors have declared that no competing interests exist.

(0.12). The significant factors influencing oral health status included age, education level, income, and access to dental services.

## Conclusion

This study provides valuable insights into the diverse oral health profiles of older adults living in Ecuador. These findings can inform targeted prevention strategies and personalized dental care interventions to improve quality of life and overall well-being.

---

## 1. Background

Projections indicate a consistent upward trajectory in the proportion of older adults within the population [1,2]. Specifically, the percentage of older adults is poised to escalate from 11.0% to 22.0% between 2000 and 2050. Furthermore, estimations suggest that by 2050, the count of dependent older adults in developing countries will undergo a fourfold increase [3]. The pervasiveness of oral diseases worldwide is on the rise, and this trend holds particularly true for older adults, who are inherently more susceptible to functional issues [4]. Recent studies conducted in Ecuador have unveiled swift transformations in the nation's demographics, health landscape, and dietary behaviors [5–8]. It is worth noting that dental issues can substantially undermine the quality of life, nutrition, and overall well-being of the older population [9–11].

Ecuador's aging population highlights the critical need for understanding the heterogeneity in oral health status among older adults. Rather than viewing this population as a homogeneous group, recognizing distinct oral health profiles can reveal specific patterns of dental problems, care needs, and intervention opportunities. Identifying these different subgroups is essential for developing targeted preventive strategies and personalized dental care interventions that address the unique challenges faced by each segment of the older adult population.

Understanding oral health patterns across Latin America provides important regional context for our study in Ecuador. Research from neighboring countries offers valuable comparative insights. Velázquez-Olmedo et al. [3] examined oral health deficits and frailty among older Mexican adults, identifying distinct classes of oral health status that significantly predicted frailty development. Their findings demonstrated how poor oral health and edentulism were associated with increased frailty risk, highlighting the broader health implications of oral conditions. Additionally, Ruiz Mendoza and Morales Borrero [28] analyzed the social determination of oral health processes across four Latin American countries, revealing common structural factors that influence oral health disparities throughout the region, including socioeconomic inequality, healthcare accessibility barriers, and varying implementation of oral health policies. These regional studies suggest that oral health challenges extend beyond national boundaries and reflect shared socioeconomic and healthcare delivery patterns across Latin America, reinforcing the importance of identifying distinct oral health profiles to guide targeted interventions in Ecuador.

Two key studies provide insights into the relationship between oral health and the development of frailty in older adults. A study from the Cohort of Obesity, Sarcopenia, and Frailty of Older Mexican Adults (COSFOMA), conducted between 2015 and 2016, examined oral health deficits as predictors of frailty [3]. The findings revealed that older adults with edentulism and poor oral health had a significantly higher risk of developing frailty compared to those with better oral health. This underscores the crucial role of oral health in maintaining the functional status and overall well-being of older adults [14]. On the other hand, a study from the United States emphasizes the importance of preventive and therapeutic dental care in older adults [15]. This approach emphasizes the need for comprehensive dental care that goes beyond emergency treatments, stressing the importance of proactive oral health management to prevent future issues. By offering a broad perspective on dental care, the study underscores how preventive strategies are essential for maintaining the overall health and quality of life of older adults [16].

Oral health in Ecuadorian older adults has traditionally been assessed through indices that may not fully capture their diverse needs. Limited research has explored how specific oral health profiles align with preventive care requirements. Del Brutto et al. [17] highlighted the association between severe edentulism and cognitive impairment in rural older adults, suggesting that certain subgroups within the older population may face compounded health challenges that extend beyond oral health. This finding points to the need for identifying distinct profiles to better understand these interconnected health issues. Additionally, Curtis et al. [18] reported significant dental decay in Ecuadorian children, indicating that oral health disparities begin early in life and may persist into older age, further supporting the need to identify different patterns of oral health status across age groups and geographical regions.

To address these gaps in understanding, this study aims to characterize the oral health profiles of older adults in Ecuador using Latent Class Analysis (LCA), an analytical approach that can identify meaningful subgroups based on patterns of responses to key oral health indicators. By focusing on variables such as tooth loss, denture usage, and the severity of dental problems, we seek to identify distinct profiles that can inform tailored prevention programs and public health strategies. This approach moves beyond traditional single-measure assessments to capture the multidimensional nature of oral health challenges in this population, enabling more effective interventions to reduce oral health disparities and address the specific needs of different subgroups [19,20]. Through this quantitative analysis, our findings will provide crucial insights for policymakers, guiding interventions to improve the quality of life and well-being of Ecuadorian older adults [21].

## 2. Methods

The study utilizes data from the Survey on Health, Well-being, and Aging (SABE), conducted in 2009 with 5,235 Ecuadorian adults aged 60 and over. The survey was coordinated by the National Institute of Statistics and Censuses (INEC) and financed by the Aliméntate Ecuador Program, the Ministry of Health, San Francisco University, and the Ministry of Economic and Social Inclusion [22]. Face-to-face interviews were carried out by trained fieldworkers, using standardized questionnaires to assess demographics, health indicators, and living conditions. Participants were randomly selected from rural and urban areas, excluding the Galápagos and Amazonia provinces [22]. While the dataset is from 2009 and certain demographic and health care aspects may have evolved, it remains the most comprehensive survey of older adult's health in Ecuador and provides valuable insights into patterns of oral health needs, which can inform current research directions and policy considerations. Moreover, the dataset has been used in various recent studies, underscoring its continued relevance for understanding health trends among older populations in the country. Ecuador is characterized by significant geographic and socioeconomic diversity across its provinces [5,6]. The country is divided into four distinct geographical regions: the coastal lowlands (Costa), the Andean highlands (Sierra), the Amazon rainforest (Oriente), and the Galápagos Islands. These regions differ considerably in terms of healthcare access, economic development, and cultural practices [5,7]. Provinces like Guayas and Pichincha, which include the major urban centers of Guayaquil and Quito respectively, have better healthcare infrastructure and higher socioeconomic indicators compared to more rural provinces [22]. In contrast, provinces such as Esmeraldas, Manabí, and those in the Amazon region typically have more

limited healthcare access, higher poverty rates, and greater challenges in maintaining oral health [7,8]. Urban areas generally offer better access to dental services, while rural communities often face barriers including distance to care facilities, financial constraints, and limited availability of specialized dental professionals [5]. These geographic and socioeconomic disparities likely influence the oral health profiles identified in this study.

Latent class analysis was employed to classify the dental health of Ecuadorian older adults. Latent Class Analysis (LCA) is a statistical method that identifies unobserved (latent) subgroups within a population based on response patterns across multiple observed variables, making it particularly suitable for identifying distinct oral health profiles [12,13,23]. The classification was based on 14 parameters: degree of edentulism, use of dentures, frequency of dental issues (such as changes in diet and difficulty chewing hard foods), ability to swallow properly, clarity of speech, discomfort while eating due to dental problems, self-consciousness in public, discomfort when looking in the mirror, use of medications to alleviate dental pain, concerns about dental status, nervousness caused by dental issues, comfort when eating in front of others, and sensitivity to cold, hot, or sweet foods.

## 2.1. Predictors of latent class membership

The analysis included a comprehensive set of demographic and risk factors related to older adults. These predictors were not derived from patients' medical records but rather from their responses to a survey.

Demographic factors included age, sex, whether they had visited a dentist in the past 12 months, and the type of residential area (urban or rural) they lived in.

As for the risk factors, they encompassed various aspects such as medication intake, overall health status, and the presence of chronic diseases like high blood pressure, diabetes, heart problems, arthritis, rheumatism or osteoarthritis, and osteoporosis. Other risk factors considered were weight, height, alcohol consumption, smoking habits, reasons for not visiting the dentist, and presence of nervous or mental problems [22].

## 2.2. Analysis

Models were fitted from one to ten classes in Stata 17.0, to identify the best-fitting latent class model, evaluated using the Akaike Information Criterion (AIC), Bayesian Information Criterion (BIC), and entropy [23]. The model selected had the lowest AIC and BIC values, along with the highest entropy [24]. After selecting the best model, a multinomial logistic regression was performed to determine significant predictors of latent class membership. [25]. The results are reported as relative risk ratios with 95% confidence intervals.

Prior to estimating the latent class models we screened the 14 indicators for collinearity. Pairwise Pearson correlations ranged from −0.09 to 0.78 (median 0.32); only two pairs—"trouble chewing hard food" with "changing meals due to teeth" (r = 0.78) and "sensitivity to temperature/sweet foods" with "discomfort when eating" (r = 0.74)—approached the 0.80 threshold often used to flag redundancy. Variance-inflation factors (VIF), obtained from an OLS regression including all indicators, averaged 2.77 (range 1.11–3.94); the corresponding tolerances (1/VIF) ranged from 0.25 to 0.90, comfortably within accepted limits (problematic if VIF > 5 or tolerance < 0.20). To confirm robustness, we re-ran the analysis after omitting each of the two most correlated items individually and together; the eight-class solution and substantive interpretations were unchanged. These diagnostics show that shared variance among indicators is modest, permitting retention of all items to preserve the full conceptual breadth of the oral-health construct without jeopardizing model stability.

Before conducting the analysis, we examined the patterns of missing data among the LCA indicator variables. We found that missing data was trivial across most indicators. Our latent class models were estimated using full information maximum likelihood estimation, which is the default approach in LCA for handling missing data. This method assumes data are Missing Completely At Random (MCAR) and allows for the inclusion of all available data from each participant rather than requiring complete cases, thus preserving sample size and statistical power while producing unbiased parameter estimates under the MCAR assumption.

Models were fitted from one to ten classes in Stata 17.0, to identify the best-fitting latent class model, evaluated using the Akaike Information Criterion (AIC), Bayesian Information Criterion (BIC), and entropy [23]. The model selected had the lowest AIC and BIC values, along with the highest entropy [24]. After selecting the best model, a multinomial logistic regression was performed to determine significant predictors of latent class membership, with Class 1 selected as the reference category due to its highest marginal probability and clinically significant characteristics [25]. The results are reported as relative risk ratios with 95% confidence intervals.

## 2.3. Ethical considerations

This study relies on data from the 2009 Survey on Health, Well-being, and Aging (SABE) conducted by the National Institute of Statistics and Censuses of Ecuador (INEC). Informed consent was obtained from all survey participants, adhering to ethical guidelines. According to Ecuadorian regulations, research using open or public data is considered low risk (Article 43, Letter b). Additionally, the Institutional Review Board (IRB) at UEES determined that, as there was no direct interaction with human subjects and the dataset is publicly available, IRB approval is not required, ensuring ethical research practices. The dataset file analysed during the current study is available from https://anda.inec.gob.ec/anda/index.php/catalog/292

## 3. Results

A comprehensive snapshot of our analytical sample is provided by Table 1. The largest demographic segment was individuals aged 60–69 years, constituting 51.37% of the sample, closely followed by those aged 70–79 years at 33.58%. Gender distribution was nearly even, with a slightly higher representation of females (52.13%) compared to males (47.87%). The majority of participants resided in urban areas (57.35%) and reported taking prescribed medication (79.99%). In terms of general health assessment, a predominant self-rating emerged as "regular" (55.53%).

Turning to oral health, participants displayed a range of concerns and denture usage patterns. More than 42.93% struggled with chewing hard food, while a substantial portion relied on dentures (65.41%). Geographically, participants were drawn from diverse provinces, with the largest contingents from Guayas (27.2%) and Pichincha (16.04%). Notably, a significant proportion of participants had not accessed dental care (75.17%), often due to financial limitations or personal hesitations. About 10.89% reported grappling with mental health issues. Shifting focus to lifestyle factors, the majority abstained from alcohol (78.36%) and never engaged in smoking (59.62%).

Comorbidity reporting was notable, revealing instances such as high blood pressure (44.33%), diabetes (13.26%), heart problems (11.89%), arthritis, rheumatism, and arthrosis (30.62%), as well as osteoporosis (17.41%). This holistic view illuminates the multifaceted nature of the participants' health profiles.

## 3.1. Latent class analysis

Fit statistics, such as Akaike Information Criterion, Bayesian Information Criterion, and entropy, were calculated, as shown in Table 2.

Table 2 provides fit statistics for models with varying numbers of classes in a Latent Class Analysis (LCA). We observed that both the Akaike Information Criterion (AIC) and Bayesian Information Criterion (BIC) gradually decreased with an increase in the number of classes, which suggests an improved model fit with greater complexity. Simultaneously, high entropy values were desirable as they indicated a high degree of accuracy in classifying individuals into their respective latent classes.

Upon a careful review of the table, we found that the eight-class model emerged as the optimal choice. It achieved lower AIC and BIC values compared to smaller models, indicating better model fit, while maintaining a high entropy value of 0.9521, indicative of clear class separation. This balance between statistical rigor and interpretability solidified our decision to adopt the eight-class model, aligning with the principle of parsimony, while offering rich insight into the heterogeneity of the analytical sample regarding the variables introduced in the model.

**Table 1. Descriptive Statistics of the Analytical Sample.**

| Variable | Percentage of Total |
|---|---|
| **Age in categories** | |
| 60 to 69 years | 51,37% |
| 70 to 79 years | 33,58% |
| 80 to 89 years | 13,30% |
| 90 years and over | 1,75% |
| **Gender** | |
| Female | 52,13% |
| Male | 47,87% |
| **Area** | |
| Urban | 57,35% |
| Rural | 42,65% |
| **Medicaments** | |
| No | 20,01% |
| Yes | 79,99% |
| **General health status** | |
| Excellent | 1,16% |
| Very good | 2,74% |
| Good | 20,72% |
| Regular | 55,53% |
| Bad | 19,85% |
| **High pressure** | |
| No | 55,67% |
| Yes | 44,33% |
| **Diabetes** | |
| No | 86,74% |
| Yes | 13,26% |
| **Heart problems** | |
| No | 88,11% |
| Yes | 11,89% |
| **Arthritis, rheumatism or arthrosis** | |
| No | 69,38% |
| Yes | 30,62% |
| **Osteoporosis** | |
| No | 82,59% |
| Yes | 17,41% |
| **Alcohol** | |
| I did not consume | 78,36% |
| Less than 1 day per week | 15,26% |
| 1 day a week | 4,02% |
| 2 to 3 days a week | 1,54% |
| 4 to 6 days a week | 0,33% |
| Every day | 0,50% |
| **Tobacco** | |
| I currently smoke | 10,30% |
| I used to smoke but not anymore | 30,08% |
| I have never smoked | 59,62% |

*(Continued)*

**Table 1.** (Continued)

| Variable | Percentage of Total |
|---|---|
| **Dentist attention** | |
| No | 75,17% |
| Yes | 24,83% |
| **Reasons for not going to the dentist** | |
| I didn't want to go | 15,55% |
| Dentist is far | 2,84% |
| I couldn't pay | 25,33% |
| I had no insurance | 0,19% |
| I have had no one to take me | 1,30% |
| Other | 29,96% |
| I have been to the dentist | 24,83% |
| **Mental diseases** | |
| No | 89,11% |
| Yes | 10,89% |
| **Province** | |
| Azuay | 5,95% |
| Bolívar | 1,51% |
| Cañar | 2,55% |
| Carchi | 1,58% |
| Cotopaxi | 3,66% |
| Chimborazo | 3,19% |
| El Oro | 4,82% |
| Esmeraldas | 2,93% |
| Guayas | 27,20% |
| Imbabura | 4,32% |
| Loja | 5,93% |
| Los Ríos | 4,02% |
| Manabí | 11,79% |
| Pichincha | 16,04% |
| Tungurahua | 4,49% |
| **Having all teeth or missing a few teeth** | |
| No | 87,52% |
| Yes | 12,48% |
| **Missing many teeth** | |
| No | 50,12% |
| Yes | 49,88% |
| **Missing all teeth** | |
| No | 62,36% |
| Yes | 37,64% |
| **Not wearing dentures** | |
| No | 66,73% |
| Yes | 33,27% |
| **Wearing dentures** | |
| No | 34,59% |
| Yes | 65,41% |

*(Continued)*

**Table 1.** (Continued)

| Variable | Percentage of Total |
| --- | --- |
| **Having full teeth** | |
| No | 98,68% |
| Yes | 1,32% |
| **Changing meals for problems with teeth** | |
| No | 72,90% |
| Yes | 27,10% |
| **Having trouble chewing hard food** | |
| No | 57,07% |
| Yes | 42,93% |
| **Having trouble speaking well** | |
| No | 82,70% |
| Yes | 17,30% |
| **Having trouble eating due to tooth discomfort** | |
| No | 68,10% |
| Yes | 31,90% |
| **Having trouble going out or talking to others** | |
| No | 88,78% |
| Yes | 11,22% |
| **Being happy when looking in the mirror** | |
| No | 40,55% |
| Yes | 59,45% |
| **Using medications to relieve dental pain** | |
| No | 86,27% |
| Yes | 13,73% |
| **Concerning about dental status** | |
| No | 59,52% |
| Yes | 40,48% |
| **Being nervous about dental problems** | |
| No | 80,17% |
| Yes | 19,83% |
| **Having trouble eating in front of others** | |
| No | 77,95% |
| Yes | 22,05% |
| **Rarely having dental discomfort due to food temperature or sweetness** | |
| No | 54,70% |
| Yes | 45,30% |
| **Frequently having dental discomfort due to food temperature or sweetness** | |
| No | 81,05% |
| Yes | 18,95% |
| **Not knowing dental discomfort due to food temperature or sweetness** | |
| No | 64,25% |
| Yes | 35,75% |

**Table 2. Fit statistics.**

| Classes | AIC | BIC | Entropy |
|---|---|---|---|
| 1 | 96,720.6 | 96,842.9 | . |
| 2 | 86,644.6 | 86,895.7 | 0.8673 |
| 3 | 75,556.4 | 75,936.3 | 0.9367 |
| 4 | 73,093.0 | 73,588.7 | 0.9391 |
| 5 | 71,191.8 | 71,829.2 | 0.9314 |
| 6 | 70,903.2 | 71,650.1 | 0.9314 |
| 7 | 67,676.8 | 68,571.7 | 0.9480 |
| 8 | 66,394.3 | 67,418.0 | 0.9521 |
| 9 | 67,581.7 | 68,734.1 | 0.9248 |
| 10 | 65,960.6 | 67,241.8 | 0.9406 |

Class 1 comprises older adults who lack all teeth, wear dentures, experience minimal dental discomfort from food temperature or sweetness, and express low consensus on general dental issues. They are less inclined to agree with statements such as 'Changing meals for problems with teeth' (PR = 0.07), 'Using medications to relieve dental pain' (PR = 0.03), and 'Being nervous about dental problems' (PR = 0.02). They typically agree with 'Being happy when looking in the mirror' (PR = 0.68). (Fig 1)

Class 2 includes older adults predominantly missing multiple teeth but rarely all, wearing dentures, and showing limited concurrence on general dental problems. They are less likely to agree with 'Changing meals for problems with teeth' (PR = 0.07) and typically agree with 'Being happy when looking in the mirror' (PR = 0.68). They strongly agree with 'Rarely having dental discomfort due to food temperature or sweetness' (PR = 0.96). (Fig 2)

Class 3 encompasses older adults with mostly intact teeth or a few missing, primarily without dentures, seldom facing difficulty in social interactions, and expressing low consensus on general dental problems. They tend to disagree with 'Having trouble chewing hard food' (PR = 0.11), 'Having trouble eating due to tooth discomfort' (PR = 0.04), 'Having trouble speaking well' (PR = 0.03), and 'Having trouble eating in front of others' (PR = 0.03). They often agree with 'Being

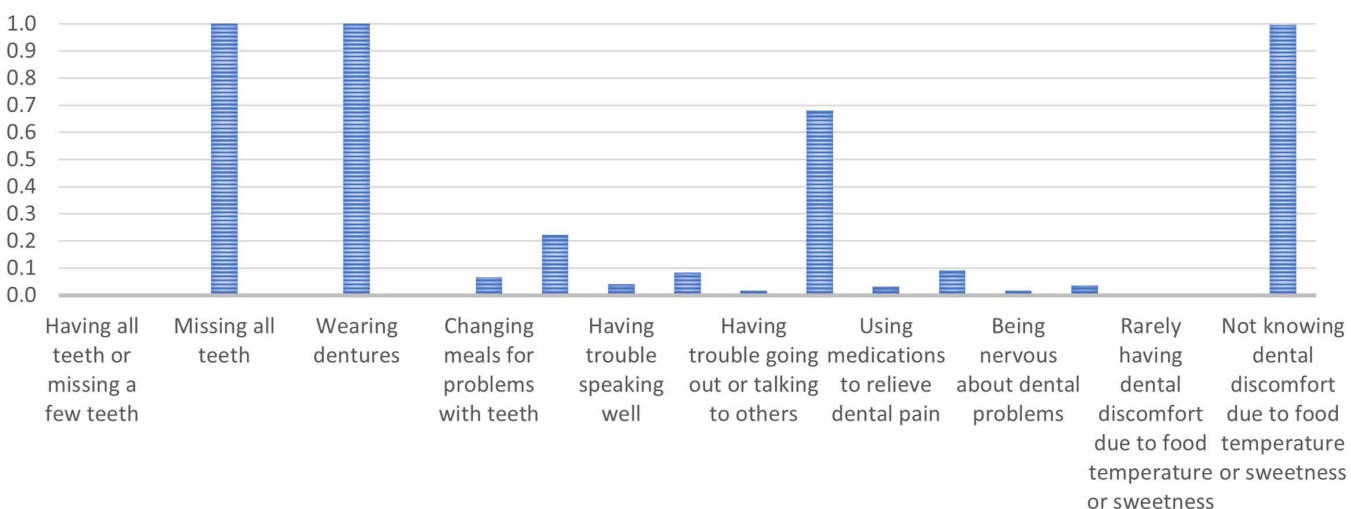

**Fig 1. Marginal probabilities for Class 1 – People with no original teeth, complete dentures, minimal problems, and high satisfaction.**

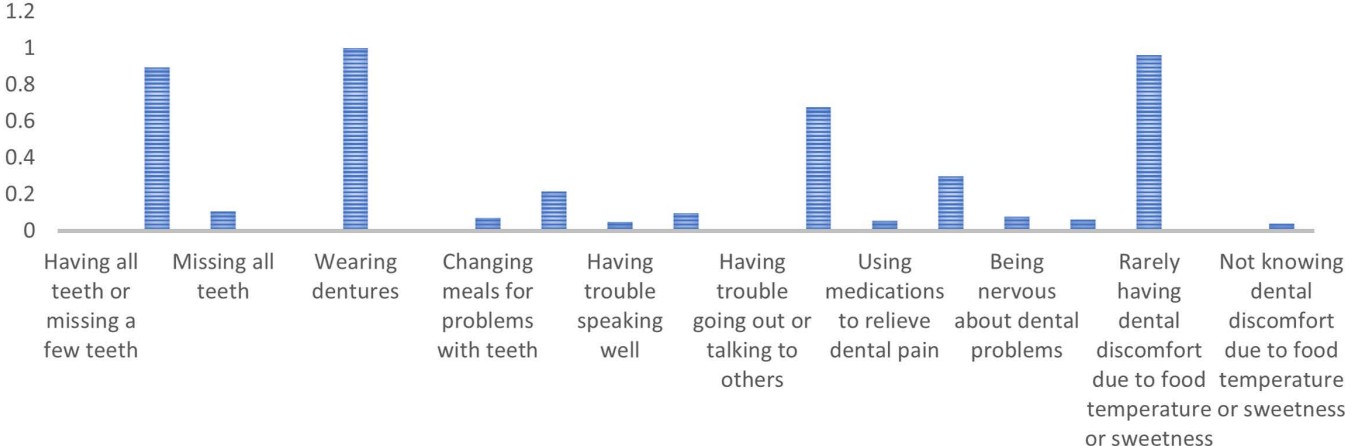

**Fig 2. Marginal probabilities for Class 2: People with missing teeth, dentures, minimal problems, and high satisfaction.**

happy when looking in the mirror' (PR = 0.72) and 'Rarely having dental discomfort due to food temperature or sweetness' (PR = 0.81). (Fig 3)

Class 4 comprises older adults mostly missing several teeth but rarely all, not using dentures, and displaying limited agreement on general dental issues. They typically agree with 'Being happy when looking in the mirror' (PR = 0.48) and strongly agree with 'Rarely having dental discomfort due to food temperature or sweetness' (PR = 0.98). (Fig 4)

Class 5 represents older adults missing numerous teeth, seldom using dentures, frequently experiencing dental discomfort from food temperature or sweetness, and holding low to moderate consensus on general dental problems. They likely agree with 'Having trouble chewing hard food' (PR = 0.55) and 'Being happy when looking in the mirror' (PR = 0.56), while also associating with 'Concerning about dental status' (PR = 0.60). (Fig 5)

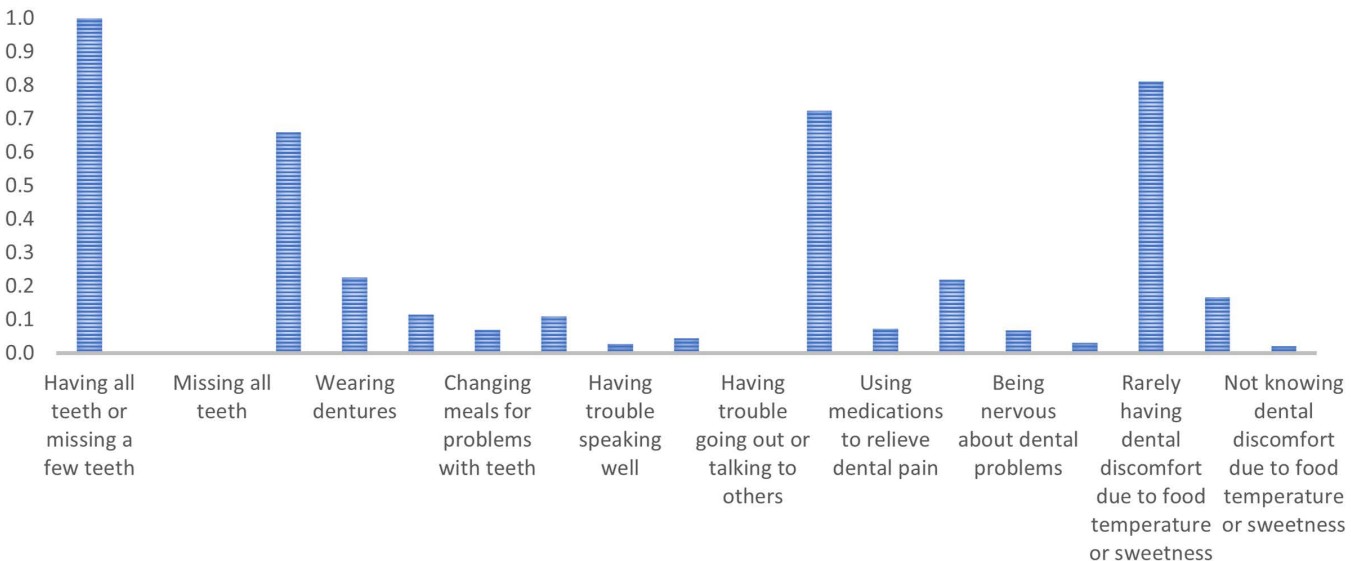

**Fig 3. Marginal probabilities for Class 3 – People with few or no missing teeth, some problems, but relatively satisfied.**

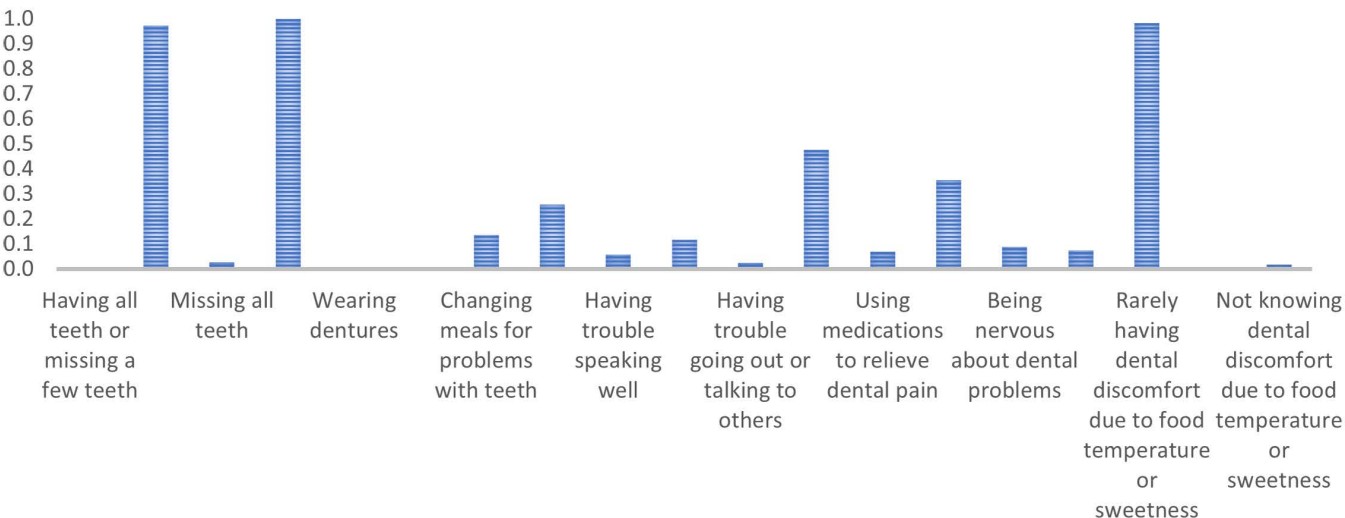

**Fig 4. Marginal probabilities for Class 4 – People with many missing teeth, no dentures, moderate problems, and high satisfaction.**

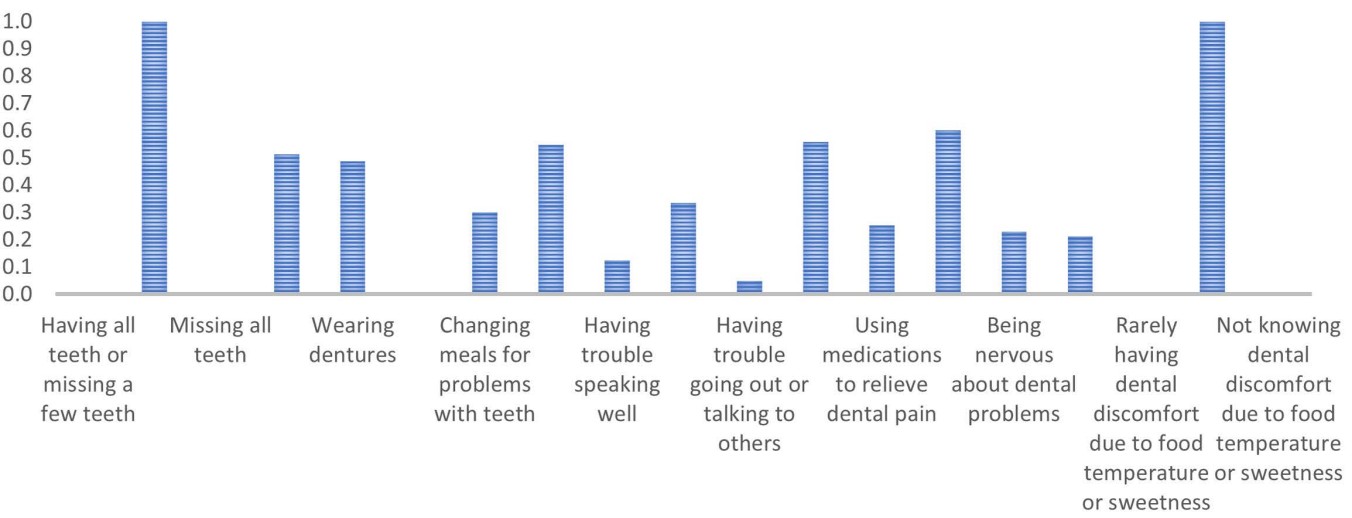

**Fig 5. Marginal probabilities for Class 5: People missing many teeth, some with dentures, moderate problems, and relatively satisfaction.**

Class 6 includes older adults who lack all teeth, predominantly use dentures, report minimal dental discomfort from food temperature or sweetness, and display low to high consensus on general dental problems. They are quite likely to agree with 'Having trouble chewing hard food' (PR = 0.86), 'Having trouble eating due to tooth discomfort' (PR = 0.77), and 'Changing meals for problems with teeth' (PR = 0.66). They tend to agree with 'Concerning about dental status' (PR = 0.63) and 'Having trouble eating in front of others' (PR = 0.50), but typically agree with 'Being happy when looking in the mirror' (PR = 0.56). (Fig 6)

Class 7 encompasses older adults largely missing multiple teeth, seldom using dentures, and displaying moderate to high consensus on general dental issues. They are most likely to agree with 'Having trouble chewing hard food' (PR = 0.94), 'Having trouble eating due to tooth discomfort' (PR = 0.82), 'Concerning about dental status' (PR = 0.74),

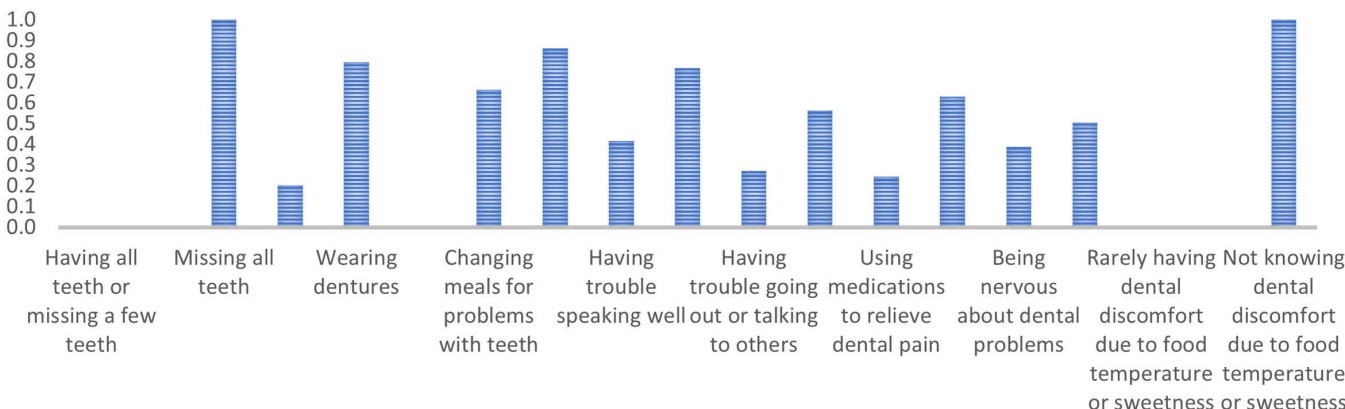

**Fig 6. Marginal probabilities for Class 6: People with all teeth missing, wearing dentures, significant problems, and high satisfaction.**

and 'Changing meals for problems with teeth' (PR = 0.67). They often agree with 'Having trouble eating in front of others' (PR = 0.52) and typically agree with 'Being happy when looking in the mirror' (PR = 0.41), while strongly agreeing with 'Rarely having dental discomfort due to food temperature or sweetness' (PR = 0.91) and rarely agreeing with 'Not knowing dental discomfort due to food temperature or sweetness' (PR = 0.09). (Fig 7)

Class 8 comprises older adults mostly missing multiple teeth, often not using dentures, frequently experiencing dental discomfort from food temperature or sweetness, and showing moderate to high consensus on general dental problems. They are most likely to agree with 'Having trouble eating due to tooth discomfort' (PR = 0.93), 'Concerning about dental status' (PR = 0.92), 'Having trouble eating in front of others' (PR = 0.77), and 'Changing meals for problems with teeth' (PR = 0.74). They also tend to strongly agree with 'Having trouble chewing hard food' (PR = 0.93) and strongly agree with 'Being nervous about dental problems' (PR = 0.69), 'Having trouble speaking well' (PR = 0.57), 'Having trouble going out or talking to others' (PR = 0.47), and 'Using medications to relieve dental pain' (PR = 0.41). However, they strongly agree with

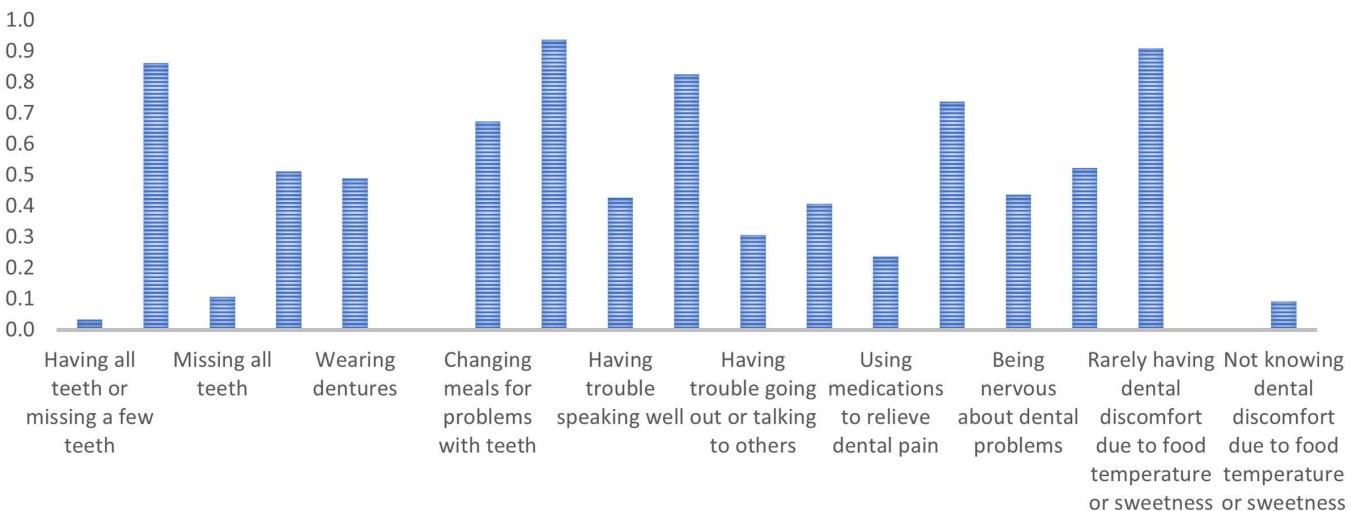

**Fig 7. Marginal probabilities for Class 7: People with many missing teeth, dentures, significant problems and relatively satisfaction.**

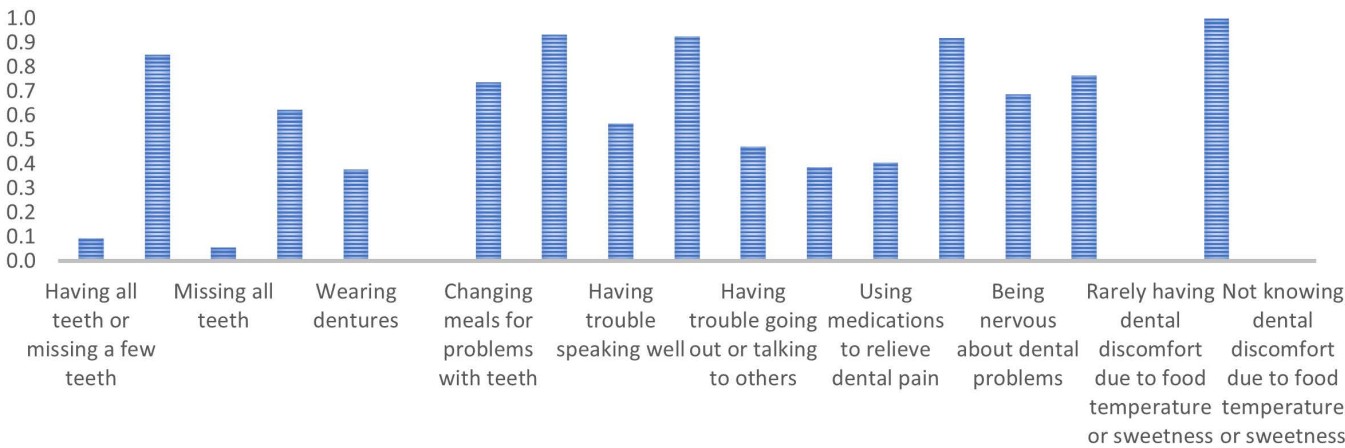

**Fig 8. Marginal probabilities for Class 8 – People with some missing teeth, no dentures, significant problems and low satisfaction.**

'Rarely having dental discomfort due to food temperature or sweetness' (PR = 0.91) and seldom agree with 'Not knowing dental discomfort due to food temperature or sweetness' (PR = 0.09). (Fig 8)

After performing the LCA, we allocated each individual to the latent class where their highest posterior probability lay. Subsequently, we employed a multinomial logistic regression model, utilizing a set of demographic and health variables, to forecast the membership of each class. The anticipated probabilities or margins for each latent class, considering the specific characteristics, are presented in Appendix 1.

In the following paragraphs, we provide a detailed class-by-class interpretation of the data presented in the Margin table (refer to Appendix 2), Appendix 3 contains the coefficients from the original multinomial logistic regression. The purpose of this interpretation was to shed light on the complex relationships between demographic and health variables and their influence on the respective latent classes of dental health. Each class is characterized by describing which groups (based on their demographic or health characteristics) are more or less likely to belong to it. We drew these conclusions by interpreting the margins or probabilities associated with each variable for each latent class. The 'more likely' groups have higher corresponding probabilities indicating a higher likelihood of being in a particular class, whereas 'less likely' groups have lower probabilities.

- **Class 1: People with no original teeth, complete dentures, minimal problems, and high satisfaction.**

More likely: People of advanced age (90 years and over, probability 0.75), those living in the Azuay (probability 0.70), Carchi (probability 0.68), or Loja (probability 0.65) provinces, people who consume alcohol every day (probability 0.64), and those with high blood pressure (probability 0.62).

Less likely: Men (probability 0.30), individuals residing in provinces such as Manabí (probability 0.25), Esmeraldas (probability 0.27), Guayas (probability 0.28), or Tungurahua (probability 0.29), and those who are in excellent or very good health (probability 0.32).

- **Class 2: People with missing teeth, dentures, minimal problems, and high satisfaction.**

More likely: Individuals residing in urban areas (probability 0.71); people who have an excellent general health status (probability 0.70); those residing in the Guayas (probability 0.69), Pichincha (probability 0.67), or El Oro (probability 0.66) provinces; and current smokers (probability 0.68). Moreover, people who consumed alcohol once a week (probability 0.65) were more likely to belong to this class.

Less likely: People in older age categories (90 years and over, probability 0.33); those residing in the Carchi (probability 0.30), Azuay (probability 0.31), or Cotopaxi (probability 0.32) provinces; and those with heart problems (probability 0.35).

• Class **3: People with few or no missing teeth and some problems, but relatively satisfied.**

More likely: Younger aged individuals (60–69 years, probability 0.73), females (probability 0.71), those residing in provinces such as Bolívar (probability 0.70) or Cañar (probability 0.67), and people who consume alcohol one day a week (probability 0.68).
    Less likely: People of advanced age (90 years and over, probability 0.34), individuals with high blood pressure (probability 0.36), those living in rural areas (probability 0.37), or residing in provinces such as Azuay (probability 0.38) or Carchi (probability 0.39).

• **Class 4: People with many missing teeth, no dentures, moderate problems, and high satisfaction.**

More likely: males (probability 0.72), those who consume alcohol two to three days a week (probability 0.70), and individuals residing in the provinces of Manabí (probability 0.69), Los Ríos (probability 0.68), or Esmeraldas (probability 0.67).
    Less likely: People with an excellent general health status (probability 0.34); those residing in the Azuay (probability 0.32), Carchi (probability 0.33), or Loja (probability 0.35) provinces; and people who do not consume alcohol (probability 0.36).

• **Class 5: People with many missing teeth, some with dentures, moderate** problems**, and relatively satisfied.**

More likely: individuals who consumed alcohol two to three days a week (probability0.74), those residing in provinces such as Imbabura (probability 0.72), Tungurahua (probability 0.70), or Santa Elena (probability 0.68), and those with mental diseases (probability 0.66).
    Less likely: Females (probability 0.33), people with excellent general health status (probability 0.35), those residing in provinces such as Carchi (probability 0.30), Azuay (probability 0.31), or Cotopaxi (probability 0.32), and those who consumed alcohol every day (probability 0.36).

• **Class 6: People with missing teeth, wearing dentures, significant** problems**, and high satisfaction.**

More likely: Males (probability 0.75), people in the 80–89 years age category (probability 0.73), individuals who consumed alcohol every day (probability 0.71), and residents of provinces such as Manabí (probability 0.69), Los Ríos (probability 0.68), and Santo Domingo de los Tsáchilas (probability 0.67).
    Less likely: Females (probability 0.30), people who are in excellent or very good health (probability 0.32), and those residing in provinces such as Guayas (probability 0.28), Pichincha (probability 0.29), or El Oro (probability 0.31).

• **Class 7: People with many missing teeth, dentures, significant problems, and relatively satisfied**

More likely: Females (probability 0.72), individuals with a history of stroke (probability 0.70), people with poor health status (probability 0.68), and those residing in provinces such as Chimborazo (probability 0.67), Sucumbíos (probability 0.65), or Zamora Chinchipe (probability 0.64).
    Less likely: Individuals with excellent or very good general health status (probability 0.33) and people living in provinces such as Guayas (probability 0.31), Pichincha (probability 0.30), or El Oro (probability 0.29).

• **Class 8: People with some missing teeth, no dentures, significant problems, and low satisfaction.**

More likely: People who consumed alcohol four to six days a week (probability 0.73), individuals who had mental diseases (probability 0.71), those who had stopped smoking (probability 0.69), and residents of provinces like Los Ríos (probability 0.67), Esmeraldas (probability 0.66), or Manabí (probability 0.65).

Less likely: individuals in the advanced age categories (90 years and over, probability 0.34), people with excellent or very good health status (probability 0.36), and those residing in provinces such as Bolívar (probability 0.33), Carchi (probability 0.31), or Azuay (probability 0.30).

## 4. Discussion

Our findings highlight the diversity of oral health profiles in older Ecuadorian adults and resonate with existing literature that underscores the multifaceted importance of oral health across different populations [18,26–28]. Del Brutto et al., for instance, found a significant association between severe edentulism and cognitive impairment in older adults from rural areas in Ecuador [17]. This emphasizes that oral health issues can have implications extending beyond mere dental well-being, potentially impacting cognitive function. These results reinforce the need to view oral health not just as a matter of dental hygiene, but also as an indicator of overall health in the older adults. On the other hand, Curtis et al. highlighted the oral health challenges faced by children in rural Ecuadorian communities, showing that concerns about oral health are not confined to older adults alone but persist throughout one's life [29]. The prevalence of dental decay in children suggests that early interventions could be beneficial in preventing more severe dental problems in adulthood. Together, these studies, along with our research, underline the pressing need for public health policies and strategies aimed at enhancing oral health across all life stages in Ecuador.

The data analysis presented key findings across the various classes. Firstly, older adults in the classes of individuals with no original teeth (Class 1), those with many missing teeth and dentures (Class 2), those with many missing teeth without dentures (Class 4), and those with all teeth missing wearing dentures with significant issues (Class 6), as well as those with many missing teeth without dentures showing high tooth sensitivity and low satisfaction (Class 8), experienced less dental discomfort associated with food temperature or sweetness. However, those in the class of individuals with many missing teeth, some with dentures and moderate issues (Class 5) and the aforementioned Class 8, frequently encountered such discomfort. Moreover, the agreement levels concerning general dental problems varied across classes, with some displaying low agreement and others indicating higher levels. Classes 1, 2, 6, and 8 were generally less inclined to alter meals due to dental issues and to resort to medications for dental pain relief. On the other hand, the classes of individuals with few or no missing teeth (Class 3), Class 5, Class 6, and those with many missing teeth, some with dentures and significant problems (Class 7), showed higher levels of concern about their dental status. Lastly, classes 1, 2, 4, 6, and 8 leaned towards agreeing on feeling content when looking in the mirror, whereas Class 7 exhibited a lower level of agreement in this regard. Collectively, these findings offer insights into the variations and similarities among classes concerning dental discomfort, denture use, agreement on general dental issues, meal alterations, dental pain relief, concern about dental status, and emotional well-being.

The observed results may be influenced by factors such as dental history, oral health status, treatment approaches, access to dental care, personal attitudes and behaviors, psychological factors, and socioeconomic and cultural factors [30]. These factors contribute to variations among the classes in terms of dental discomfort, denture usage, agreement about dental problems, and emotional well-being [31]. These findings suggest that individual oral health experiences and perceptions are shaped by a combination of personal, clinical, and sociocultural factors [32].

When interpreting the oral health profiles identified in this study, it may be important to consider the cultural context that potentially shapes oral health practices and perceptions in Ecuador and neighboring Andean countries. Cultural factors could influence dental care decisions, including aesthetic preferences that might differ from conventional clinical standards. For instance, in some communities throughout the region, gold inlays and visible dental modifications might be considered status symbols rather than purely restorative interventions. Such cultural preferences may affect how individuals perceive their oral health status, their satisfaction with dental prostheses, and their decisions regarding dental care. Additionally, traditional beliefs about oral health and healing practices could coexist with modern dentistry, particularly in rural and indigenous communities. These cultural dimensions might contribute to the diverse satisfaction levels observed

across different oral health profiles and could be considered when interpreting care-seeking behaviors among older Ecuadorians.

Economic constraints appear to play an important role in shaping access to dental care among older Ecuadorians. Our findings revealed that a significant proportion of participants (25.33%) cited inability to pay as the main reason for not visiting a dentist. This economic barrier may be particularly challenging in contexts where out-of-pocket healthcare expenses could represent a substantial financial burden for older adults, many of whom might rely on limited retirement incomes or family support. While public dental services are available, they may face resource limitations that could affect quality and availability, especially in rural areas. Private dental care might remain unaffordable for many older adults, potentially leading to postponed treatment and possibly worse oral health outcomes. These economic barriers could contribute to the disparities observed across the different oral health profiles identified in our study, with individuals from lower socioeconomic backgrounds possibly overrepresented in profiles characterized by missing teeth without proper prosthetic replacement. Addressing these economic considerations might be an important component of improving oral health equity among older Ecuadorians.

In research pertinent to our study, Velázquez-Olmedo et al. delved into the relationship between oral health and quality of life in older adults [3]. Their analysis yielded a three-class model, featuring edentulism, and two additional classes characterized by varying degrees of acceptable oral health and poor oral health. Specifically, they found that 6.9% of participants were edentulous, 57.9% belonged to Class 1 with acceptable oral health, 13.9% to Class 2 with acceptable oral health, and 21.3% to Class 3 with poor oral health. Notably, they observed that 18.0% of participants developed frailty and that those with edentulism or poor oral health (Class 3) had an increased risk of developing it compared to those with acceptable oral health (Class 1). These findings from Velázquez-Olmedo et al. underscore the significance of oral health in the quality of life and well-being of older adults [3]. While their study and ours adopt different methodological approaches and categorizations, both pieces of research converge on the centrality of oral health to the well-being of older adults. While their investigation highlights the risks associated with edentulism and poor oral health, our analysis offers a more detailed understanding of oral health profiles, identifying eight distinct categories among older adults in Ecuador. These results emphasize the importance of oral health in the overall well-being and frailty status of older adults [33]. This suggests that edentulism and poor oral health are associated with an elevated risk of frailty [34]. Together, these studies emphasize the need for targeted and preventive interventions in the realm of oral health.

Additionally, Sischo and Broder discussed the importance of Oral Health-related Quality of Life (OHRQoL) as we move to patient-oriented outcomes and approaches to measure treatment needs and efficiency of care [16]. These studies further validated the importance of our main results, as this research can be used to inform public policy and help eradicate oral health disparities based on the variations and similarities among different classes of older adults regarding dental discomfort, denture usage, agreement about dental problems, meal changes, dental pain relief, concern about dental status, and emotional well-being [35–39].

However, it is important to acknowledge that the findings of this research are restricted to patients' responses to a survey, and the study lacks clinical indicators. No direct clinical examination was conducted to confirm participants' dental conditions. The decision to utilize this database was necessitated by the fact that it is the most recent database in Ecuador that includes relevant dental and oral health-related questions. It is important to acknowledge that this study utilized data from 2009, and some demographic, healthcare access, and oral health patterns may have changed in the intervening years. However, the findings maintain relevance for several key reasons. As noted by Beard et al. [1] and Christensen et al. [2], global aging trends continue to increase the proportion of older adults in populations worldwide, including in Ecuador, making these findings increasingly pertinent. The oral health challenges identified in our study reflect underlying patterns that connect to broader healthcare accessibility issues and socioeconomic factors previously documented in Ecuador [5–8]. While specific prevalence rates and the exact distribution of oral health profiles may have evolved,

the fundamental oral health relationships and underlying socioeconomic disparities identified in this study likely remain relevant. As emphasized by Watt et al. [19], understanding distinct patterns of oral health needs is essential for developing targeted interventions, regardless of when the baseline data was collected. The findings should therefore be interpreted as a valuable baseline for understanding oral health patterns among older Ecuadorians, with recognition that contemporary validation would strengthen their current applicability. Future studies using more recent data would be valuable to assess how these patterns have changed over time. h.

Furthermore, incorporating additional clinical parameters would provide a more comprehensive measure of oral health deficits and indices [40]. The demand for research on this topic within this population supports the need for further investigation. However, future studies should consider clinical assessments and incorporate more comprehensive oral health parameters to enhance our understanding of oral health issues in this population [41]. This would enable a more well-rounded evaluation of the older adult's oral health, bridging the gap between self-reported perceptions and objective clinical measures.

## 5. Conclusion

In this study, our objective to identify and characterize oral health profiles in older adults in Ecuador through latent class analysis was successfully achieved, revealing eight distinct profiles based on factors like tooth loss, denture usage, and dental problems. These findings emphasize the need for tailored approaches to address the diverse oral health needs of the older adult population, with the potential to significantly enhance their overall quality of life and well-being.

These distinct oral health profiles have potential implications for public health strategies and dental care approaches in Ecuador. The identification of eight different profiles suggests that tailored interventions, rather than uniform approaches, might better address the diverse oral health needs of older adults. For instance, profiles characterized by complete tooth loss with well-functioning dentures and high satisfaction could benefit from maintenance-focused programs, while profiles showing partial edentulism without prosthetic replacement and significant functional limitations might require more comprehensive rehabilitative interventions. Health planners could consider developing targeted educational and preventive programs for specific geographic regions where certain profiles predominate. Additionally, the socioeconomic factors influencing profile membership suggest that financial support mechanisms or subsidized services might help address barriers to care among vulnerable groups. Healthcare provider training could incorporate awareness of these diverse profiles to promote more personalized care approaches. By recognizing and responding to these varied oral health patterns, policymakers may be better positioned to develop effective strategies that improve quality of life and well-being among older Ecuadorians while optimizing healthcare resource allocation.

However, it's essential to acknowledge the limitations, as the results rely on survey responses without clinical indicators, and the use of an outdated database could introduce biases. Therefore, future research incorporating clinical assessments and assessing the effectiveness of dental care programs tailored to the identified profiles is recommended, contributing to a deeper understanding of oral health needs in the older adults and improving healthcare delivery for this group.

## Supporting information

**Appendix 1. Predicted Probabilities of Classes Based on Multinomial Logistic Regression.**
(DOCX)

**Appendix 2. Marginal probabilities by each class.**
(DOCX)

**Appendix 3. Multinomial Logistic Regression.**
(DOCX)

## Author contributions

**Conceptualization:** Adriana Canessa-Rojas, Marite Falquez-Flor.

**Formal analysis:** Adriana Canessa-Rojas, Marite Falquez-Flor, Stephanie Gallegos-Caamñano, Marco Faytong-Haro.

**Methodology:** Adriana Canessa-Rojas, Marite Falquez-Flor.

**Project administration:** Marco Faytong-Haro.

**Supervision:** Marco Faytong-Haro.

**Visualization:** Adriana Canessa-Rojas, Marite Falquez-Flor.

**Writing – original draft:** Adriana Canessa-Rojas, Marite Falquez-Flor, Marco Faytong-Haro.

**Writing – review & editing:** Adriana Canessa-Rojas, Marite Falquez-Flor, Marco Faytong-Haro.

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
