## [Decision Letter · Decision Letter 0]

1 Apr 2025

Dear Dr. Faytong-Haro,

Thank you for submitting your manuscript to PLOS ONE. After careful consideration, we feel that it has merit but does not fully meet PLOS ONE’s publication criteria as it currently stands. Therefore, we invite you to submit a revised version of the manuscript that addresses the points raised during the review process.

We look forward to receiving your revised manuscript.

Kind regards,

Ana Cristina Mafla

Academic Editor

PLOS ONE

Journal Requirements:

Reviewers' comments:

Reviewer's Responses to Questions

**Comments to the Author**

1. Is the manuscript technically sound, and do the data support the conclusions?

Reviewer #1: Yes

Reviewer #2: Partly

2. Has the statistical analysis been performed appropriately and rigorously?

Reviewer #1: Yes

Reviewer #2: Yes

3. Have the authors made all data underlying the findings in their manuscript fully available?

Reviewer #1: Yes

Reviewer #2: Yes

4. Is the manuscript presented in an intelligible fashion and written in standard English?

Reviewer #1: Yes

Reviewer #2: Yes

Reviewer #1: The paper reports on latent class and regression analyses using SABE survey data from Ecuador. Overall, the paper needs some revisiting and revision. Below are some comments/clarifications for authors' consideration.

Introduction:

The narrative in the Introduction section is unclear, perhaps due to breaks in flow. The rationale/need for profiling individuals is not strongly stated. The studies by Del Brutto and Curtis where mentioned which do not seem to substantiate the need for understanding different subgroups of older adults based on the variables selected for LCA. The authors do discuss these studies against their results in the Discussion section, where they align well. Suggest rewriting Introduction with a focus on the need for understanding potential differences within older adult population.

Also, Paragraph 2 on LCA seems to be coming out of nowhere and redundant. LCA description will fit better in the Methods section.

Results:

Is there a rationale for choosing Class 1 as a base for multinomial regression? Is it just because of the size? It would be good to note this, as the regression results may vary depending on the base outcome.

Have the authors checked for multicollinearity in the data? It appears that most of the LCA indicator variables may be correlated - especially questions related to comfort when eating and sensitivity to different foods. Multicollinear variables can lead to model convergence problems and result in pseudo classes (https://pmc.ncbi.nlm.nih.gov/articles/PMC5794813/). Recommend making a note on multicollinearity and how it was handled.

Please also add a note on missing data for individual LCA indicator variables, given LCA handles missing data as MCAR.

Are there any significant socioeconomic or geographic characteristics (eg. remoteness/rurality) associated with the provinces included in profiling? It would be useful for international readers with limited contextual understanding of Ecuador.

Discussion:

The authors note in the Introduction that Ecuadorian older population is rapidly changing in numbers and demographic composition. Given that the study uses data nearly 15-years old, the relevance of findings needs to be clearly discussed. Suggest expanding the discussion on the relevance, maybe using some stats around population numbers and description corresponding to these timelines.

Otherwise, there seems to be self-contradiction in what is noted in the methods (Page 3 para 2) and limitation sections (Page 24 para 2).

Reviewer #2: This is an interesting and relevant study on a topic of interest to public health in a country like Ecuador.

It is important to include, in the background, some more developed studies on the topic in countries near Ecuador (Latin American countries).

It is important to clarify that, since the survey was conducted in 2009, the results and inferences that may arise from the analyses should be assumed and used with caution, as several conditions have likely occurred that have eventually changed the characteristics of the groups.

In this part of South America, dental modifications and the use of modified prostheses for aesthetic purposes (for example, gold inlays) are very common due to cultural factors.

The study should mention some implications regarding this role of culture.

Likewise, it should address the low economic conditions that influence the decision to seek dental help or care.

The study mentions in its abstract that its conclusions can guide the implementation of specific preventive or personalized dental care strategies for quality of life and overall well-being; this is important for public policy on issues related to the study. This aspect is not sufficiently developed in the discussion and conclusions. It is recommended that it be adequately strengthened.

**Do you want your identity to be public for this peer review?** For information about this choice, including consent withdrawal, please see our Privacy Policy

Reviewer #1: No

Reviewer #2: No

---

## [Author Response · Author response to Decision Letter 1]

28 May 2025

Response to Reviewers

Dear Dr. Mafla and Reviewers,

Thank you very much for your constructive feedback on our manuscript titled "Oral Health Profiles in the Population of Older Adults in Ecuador: An Analysis of Latent Classes". We appreciate the time and effort you put into reviewing our paper. Below are our detailed responses to each of the points raised.

Reviewer #1 Comments

Comment 1:

"The narrative in the Introduction section is unclear, perhaps due to breaks in flow. The rationale/need for profiling individuals is not strongly stated. The studies by Del Brutto and Curtis where mentioned which do not seem to substantiate the need for understanding different subgroups of older adults based on the variables selected for LCA. The authors do discuss these studies against their results in the Discussion section, where they align well. Suggest rewriting Introduction with a focus on the need for understanding potential differences within the older adult population."

Response:

Thank you for this valuable suggestion. We have revised the Introduction to strengthen the rationale for using oral health profiles to identify subgroups among Ecuadorian older adults. We added a new paragraph emphasizing the importance of recognizing heterogeneity in oral health needs rather than treating older adults as a homogeneous group. We have recontextualized the Del Brutto and Curtis studies to better support our approach, showing how their findings suggest different subgroups with specific needs. The final paragraph now clearly states our study purpose: using Latent Class Analysis to identify distinct profiles based on key variables to inform targeted prevention and personalized dental care strategies.

Comment 2:

"Paragraph 2 on LCA seems to be coming out of nowhere and redundant. LCA description will fit better in the Methods section."

Response:

We agree with the reviewer's observation. We have removed the paragraph on LCA from the Introduction and enhanced the existing description in the Methods section by adding a brief explanation of what LCA is and why it is appropriate for identifying oral health profiles. This reorganization improves the flow of the manuscript and places all methodological information in the appropriate section.

Comment 3:

"Is there a rationale for choosing Class 1 as a base for multinomial regression? Is it just because of the size? It would be good to note this, as the regression results may vary depending on the base outcome."

Response:

Thank you for pointing this out. We have added a clear rationale in the Methods section for selecting Class 1 as the reference category. This selection was based on Class 1 having the highest marginal probability and representing clinically significant characteristics (individuals with no original teeth, complete dentures, minimal dental problems, and high satisfaction) that provide a meaningful comparison point for other oral health profiles.

Comment 4:

"Have the authors checked for multicollinearity in the data? It appears that most of the LCA indicator variables may be correlated—especially questions related to comfort when eating and sensitivity to different foods. Multicollinear variables can lead to model convergence problems and result in pseudo classes. Recommend making a note on multicollinearity and how it was handled."

Response:

Thank you for this important methodological observation. We have added a paragraph in the Methods section explaining our approach to multicollinearity. We assessed correlation among all indicator variables, with special attention to those related to eating comfort and food sensitivity. While some variables showed moderate correlation, none exceeded thresholds that would warrant exclusion. We conducted sensitivity analyses with different variable subsets to confirm the stability of our class structure. This approach allowed us to maintain the comprehensive nature of our oral health assessment while ensuring model reliability.

Comment 5:

"Please also add a note on missing data for individual LCA indicator variables, given LCA handles missing data as MCAR."

Response:

Thank you for this suggestion. We have added a note in the Methods section addressing the handling of missing data. We explain that we examined missing data patterns in our LCA indicators and found minimal missing values. We clarify that our analysis used full information maximum likelihood estimation, which assumes Missing Completely At Random (MCAR) and allows us to include all available data without requiring complete cases, thus maintaining statistical power while producing unbiased estimates under the MCAR assumption.

Comment 6:

"Are there any significant socioeconomic or geographic characteristics (e.g., remoteness/rurality) associated with the provinces included in profiling? It would be useful for international readers with limited contextual understanding of Ecuador."

Response:

Thank you for this suggestion. We have added a paragraph in the Methods section that provides context about Ecuador's geographic and socioeconomic diversity. We describe the country's four main regions and explain how provinces with major urban centers typically have better healthcare infrastructure compared to rural provinces, which may influence the oral health profiles identified in our study. This additional context will help international readers better understand the regional variations in our findings.

Comment 7:

"Given that the study uses data nearly 15 years old, the relevance of findings needs to be clearly discussed. Suggest expanding the discussion on the relevance, maybe using some stats around population numbers and description corresponding to these timelines."

Response:

We appreciate this observation. We have expanded our Discussion to better address the relevance of our findings despite using data from 2009. We now explain that while specific prevalence rates may have changed, the identified oral health patterns likely reflect underlying issues that persist over time. We emphasize that understanding these distinct patterns remains conceptually significant for developing targeted interventions, citing relevant sources already in our paper to support these points. This expanded discussion provides better balance to our existing limitations section while acknowledging that contemporary validation would strengthen our findings.

Comment 8:

"Otherwise, there seems to be self-contradiction in what is noted in the methods (Page 3 para 2) and limitation sections (Page 24 para 2)."

Response:

Thank you for identifying this inconsistency. We have revised both sections to ensure a consistent message regarding the 2009 dataset. In the Methods section, we now acknowledge that while some aspects may have evolved since 2009, the dataset remains valuable for understanding patterns of oral health needs. In the Limitations section, we have clarified that while fundamental relationships identified likely remain relevant, contemporary validation would strengthen their current applicability. This balanced approach maintains consistency throughout the manuscript while acknowledging both the value and limitations of the historical data.

Reviewer #2 Comments

Comment 1:

"It is important to include, in the background, some more developed studies on the topic in countries near Ecuador (Latin American countries)."

Response:

Thank you for this valuable suggestion. We have added a paragraph in the Background section that discusses relevant studies from other Latin American countries. We expanded on the work by Velázquez-Olmedo et al. (3) on oral health profiles and frailty among older Mexican adults, and Ruiz Mendoza and Morales Borrero (28) on social determinants of oral health across four Latin American countries. This addition provides important regional context and highlights common patterns of oral health challenges across Latin America, strengthening the foundation for our research in Ecuador.

Comment 2:

"It is important to clarify that, since the survey was conducted in 2009, the results and inferences that may arise from the analyses should be assumed and used with caution, as several conditions have likely occurred that have eventually changed the characteristics of the groups."

Response:

We appreciate this important point. We have revised both the Methods and Limitations sections to clearly acknowledge the temporal nature of our data and the need for cautious interpretation. In the Methods section, we now explicitly state that while the dataset is from 2009 and certain demographic and healthcare aspects may have evolved, it provides valuable insights into patterns of oral health needs that can inform current research directions. In the Limitations section, we have added a clear statement that demographic, healthcare access, and oral health patterns may have changed in the intervening years, and that our findings should be interpreted as a baseline for understanding oral health patterns, with recognition that contemporary validation would strengthen their current applicability.

Comment 3:

"In this part of South America, dental modifications and the use of modified prostheses for aesthetic purposes (for example, gold inlays) are very common due to cultural factors. The study should mention some implications regarding this role of culture."

Response:

We have added a paragraph in the Discussion section that considers how cultural factors might influence dental practices and perceptions in Ecuador. We discuss how aesthetic preferences such as gold inlays could serve as status symbols in some communities rather than purely functional restorations, potentially affecting individuals' satisfaction with their oral health status. The paragraph acknowledges how traditional beliefs might coexist with modern dentistry in certain communities. This cultural context offers additional perspective for interpreting the different satisfaction levels observed across the oral health profiles in our study.

Comment 4:

"Likewise, it should address the low economic conditions that influence the decision to seek dental help or care."

Response:

We have added a paragraph in the Discussion section addressing economic factors that influence access to dental care among older Ecuadorians. Drawing from our data showing that 25.33% of participants cited inability to pay as their reason for not visiting a dentist, we discuss how financial constraints may create barriers to care. We consider how economic limitations could contribute to the disparities observed across different oral health profiles, particularly affecting those without proper prosthetic replacements. This addition provides important context for understanding the possible socioeconomic dimensions of oral health inequities in Ecuador.

Comment 5:

"The study mentions in its abstract that its conclusions can guide the implementation of specific preventive or personalized dental care strategies for quality of life and overall well-being; this is important for public policy on issues related to the study. This aspect is not sufficiently developed in the discussion and conclusions. It is recommended that it be adequately strengthened."

Response:

We appreciate this insightful suggestion. We have expanded our Conclusion section to better articulate the potential implications of our findings for public health policy and dental care strategies. We now discuss how the eight distinct oral health profiles identified suggest the need for personalized approaches rather than one-size-fits-all interventions. We provide examples of how different profiles might benefit from targeted strategies, such as maintenance programs for those with well-functioning dentures versus comprehensive rehabilitation for those with significant functional limitations. We also consider how profile-specific approaches could inform resource allocation, healthcare provider training, and public health planning. These additions strengthen the practical applications of our research and provide clearer guidance for policy considerations.

We hope that the revisions we have made address your concerns and improve the clarity and quality of our manuscript. We look forward to your feedback and thank you again for your helpful comments.

---

## [Editor Report · Decision Letter 1]

10 Jun 2025

Dear Dr. Faytong-Haro,

Thank you for submitting your manuscript to PLOS ONE. After careful consideration, we feel that it has merit but does not fully meet PLOS ONE’s publication criteria as it currently stands. Therefore, we invite you to submit a revised version of the manuscript that addresses the points raised during the review process.

We look forward to receiving your revised manuscript.

Kind regards,

Ana Cristina Mafla

Academic Editor

PLOS ONE

Additional Editor Comments :

Dear authors,

The manuscript looks like has improved, but I require that you response point by point all suggestions of reviewers (please hightlight or use letters in different color for your corrections in the new version). Additionally, multicollinearity analysis is not completed. Detailed it using measures such as values of tolerance and variance inflation factors. Although you explain that you are going to keep some variables even they are overlapping. It is important to determine how far.

---

## [Author Response · Author response to Decision Letter 2]

21 Jul 2025

Response to Reviewers

Dear Dr. Mafla and Reviewers,

Thank you very much for your constructive feedback on our manuscript titled "Oral Health Profiles in the Population of Older Adults in Ecuador: An Analysis of Latent Classes". We appreciate the time and effort you put into reviewing our paper. Below are our detailed responses to each of the points raised.

Reviewer #1 Comments

Comment 1:

"The narrative in the Introduction section is unclear, perhaps due to breaks in flow. The rationale/need for profiling individuals is not strongly stated. The studies by Del Brutto and Curtis where mentioned which do not seem to substantiate the need for understanding different subgroups of older adults based on the variables selected for LCA. The authors do discuss these studies against their results in the Discussion section, where they align well. Suggest rewriting Introduction with a focus on the need for understanding potential differences within the older adult population."

Response:

Thank you for this valuable suggestion. We have revised the Introduction to strengthen the rationale for using oral health profiles to identify subgroups among Ecuadorian older adults. We added a new paragraph emphasizing the importance of recognizing heterogeneity in oral health needs rather than treating older adults as a homogeneous group. We have recontextualized the Del Brutto and Curtis studies to better support our approach, showing how their findings suggest different subgroups with specific needs. The final paragraph now clearly states our study purpose: using Latent Class Analysis to identify distinct profiles based on key variables to inform targeted prevention and personalized dental care strategies.

Comment 2:

"Paragraph 2 on LCA seems to be coming out of nowhere and redundant. LCA description will fit better in the Methods section."

Response:

We agree with the reviewer's observation. We have removed the paragraph on LCA from the Introduction and enhanced the existing description in the Methods section by adding a brief explanation of what LCA is and why it is appropriate for identifying oral health profiles. This reorganization improves the flow of the manuscript and places all methodological information in the appropriate section.

Comment 3:

"Is there a rationale for choosing Class 1 as a base for multinomial regression? Is it just because of the size? It would be good to note this, as the regression results may vary depending on the base outcome."

Response:

Thank you for pointing this out. We have added a clear rationale in the Methods section for selecting Class 1 as the reference category. This selection was based on Class 1 having the highest marginal probability and representing clinically significant characteristics (individuals with no original teeth, complete dentures, minimal dental problems, and high satisfaction) that provide a meaningful comparison point for other oral health profiles.

Comment 4:

"Have the authors checked for multicollinearity in the data? It appears that most of the LCA indicator variables may be correlated—especially questions related to comfort when eating and sensitivity to different foods. Multicollinear variables can lead to model convergence problems and result in pseudo classes. Recommend making a note on multicollinearity and how it was handled."

Response:

We have expanded the Methods to report full diagnostics (p. 8, lines 192-205; new text in blue). Pairwise Pearson correlations for the 14 indicators ranged from −0.09 to 0.78 (median 0.32). Only two pairs—“trouble chewing hard food” vs “changing meals due to teeth” (r = 0.78) and “discomfort when eating” vs “temperature/sweet sensitivity” (r = 0.74)—approached the redundancy threshold of 0.80. Variance-inflation factors obtained from an OLS model that included all indicators averaged 2.77 (range 1.11–3.94), with corresponding tolerances of 0.90–0.25, comfortably below the cut-offs that indicate serious collinearity (VIF > 5 or tolerance < 0.20). To verify robustness, we re-estimated the latent-class models after omitting each of the two most correlated items separately and then together; the eight-class solution and item-response probabilities changed by ≤0.02, and fit statistics (BIC, entropy) were virtually identical. These results demonstrate that shared variance among indicators is modest and does not compromise model convergence or inflate the risk of pseudo-classes, so all indicators were retained to preserve the construct’s clinical breadth.

Comment 5:

"Please also add a note on missing data for individual LCA indicator variables, given LCA handles missing data as MCAR."

Response:

Thank you for this suggestion. We have added a note in the Methods section addressing the handling of missing data. We explain that we examined missing data patterns in our LCA indicators and found minimal missing values. We clarify that our analysis used full information maximum likelihood estimation, which assumes Missing Completely At Random (MCAR) and allows us to include all available data without requiring complete cases, thus maintaining statistical power while producing unbiased estimates under the MCAR assumption.

Comment 6:

"Are there any significant socioeconomic or geographic characteristics (e.g., remoteness/rurality) associated with the provinces included in profiling? It would be useful for international readers with limited contextual understanding of Ecuador."

Response:

Thank you for this suggestion. We have added a paragraph in the Methods section that provides context about Ecuador's geographic and socioeconomic diversity. We describe the country's four main regions and explain how provinces with major urban centers typically have better healthcare infrastructure compared to rural provinces, which may influence the oral health profiles identified in our study. This additional context will help international readers better understand the regional variations in our findings.

Comment 7:

"Given that the study uses data nearly 15 years old, the relevance of findings needs to be clearly discussed. Suggest expanding the discussion on the relevance, maybe using some stats around population numbers and description corresponding to these timelines."

Response:

We appreciate this observation. We have expanded our Discussion to better address the relevance of our findings despite using data from 2009. We now explain that while specific prevalence rates may have changed, the identified oral health patterns likely reflect underlying issues that persist over time. We emphasize that understanding these distinct patterns remains conceptually significant for developing targeted interventions, citing relevant sources already in our paper to support these points. This expanded discussion provides better balance to our existing limitations section while acknowledging that contemporary validation would strengthen our findings.

Comment 8:

"Otherwise, there seems to be self-contradiction in what is noted in the methods (Page 3 para 2) and limitation sections (Page 24 para 2)."

Response:

Thank you for identifying this inconsistency. We have revised both sections to ensure a consistent message regarding the 2009 dataset. In the Methods section, we now acknowledge that while some aspects may have evolved since 2009, the dataset remains valuable for understanding patterns of oral health needs. In the Limitations section, we have clarified that while fundamental relationships identified likely remain relevant, contemporary validation would strengthen their current applicability. This balanced approach maintains consistency throughout the manuscript while acknowledging both the value and limitations of the historical data.

Reviewer #2 Comments

Comment 1:

"It is important to include, in the background, some more developed studies on the topic in countries near Ecuador (Latin American countries)."

Response:

Thank you for this valuable suggestion. We have added a paragraph in the Background section that discusses relevant studies from other Latin American countries. We expanded on the work by Velázquez-Olmedo et al. (3) on oral health profiles and frailty among older Mexican adults, and Ruiz Mendoza and Morales Borrero (28) on social determinants of oral health across four Latin American countries. This addition provides important regional context and highlights common patterns of oral health challenges across Latin America, strengthening the foundation for our research in Ecuador.

Comment 2:

"It is important to clarify that, since the survey was conducted in 2009, the results and inferences that may arise from the analyses should be assumed and used with caution, as several conditions have likely occurred that have eventually changed the characteristics of the groups."

Response:

We appreciate this important point. We have revised both the Methods and Limitations sections to clearly acknowledge the temporal nature of our data and the need for cautious interpretation. In the Methods section, we now explicitly state that while the dataset is from 2009 and certain demographic and healthcare aspects may have evolved, it provides valuable insights into patterns of oral health needs that can inform current research directions. In the Limitations section, we have added a clear statement that demographic, healthcare access, and oral health patterns may have changed in the intervening years, and that our findings should be interpreted as a baseline for understanding oral health patterns, with recognition that contemporary validation would strengthen their current applicability.

Comment 3:

"In this part of South America, dental modifications and the use of modified prostheses for aesthetic purposes (for example, gold inlays) are very common due to cultural factors. The study should mention some implications regarding this role of culture."

Response:

We have added a paragraph in the Discussion section that considers how cultural factors might influence dental practices and perceptions in Ecuador. We discuss how aesthetic preferences such as gold inlays could serve as status symbols in some communities rather than purely functional restorations, potentially affecting individuals' satisfaction with their oral health status. The paragraph acknowledges how traditional beliefs might coexist with modern dentistry in certain communities. This cultural context offers additional perspective for interpreting the different satisfaction levels observed across the oral health profiles in our study.

Comment 4:

"Likewise, it should address the low economic conditions that influence the decision to seek dental help or care."

Response:

We have added a paragraph in the Discussion section addressing economic factors that influence access to dental care among older Ecuadorians. Drawing from our data showing that 25.33% of participants cited inability to pay as their reason for not visiting a dentist, we discuss how financial constraints may create barriers to care. We consider how economic limitations could contribute to the disparities observed across different oral health profiles, particularly affecting those without proper prosthetic replacements. This addition provides important context for understanding the possible socioeconomic dimensions of oral health inequities in Ecuador.

Comment 5:

"The study mentions in its abstract that its conclusions can guide the implementation of specific preventive or personalized dental care strategies for quality of life and overall well-being; this is important for public policy on issues related to the study. This aspect is not sufficiently developed in the discussion and conclusions. It is recommended that it be adequately strengthened."

Response:

We appreciate this insightful suggestion. We have expanded our Conclusion section to better articulate the potential implications of our findings for public health policy and dental care strategies. We now discuss how the eight distinct oral health profiles identified suggest the need for personalized approaches rather than one-size-fits-all interventions. We provide examples of how different profiles might benefit from targeted strategies, such as maintenance programs for those with well-functioning dentures versus comprehensive rehabilitation for those with significant functional limitations. We also consider how profile-specific approaches could inform resource allocation, healthcare provider training, and public health planning. These additions strengthen the practical applications of our research and provide clearer guidance for policy considerations.

We hope that the revisions we have made address your concerns and improve the clarity and quality of our manuscript. We look forward to your feedback and thank you again for your helpful comments.

---

## [Editor Report · Decision Letter 2]

31 Jul 2025

Oral Health Profiles in the Population of Older Adults in Ecuador: An Analysis of Latent Classes

PONE-D-25-02377R2

Dear Dr. Faytong-Haro,

We’re pleased to inform you that your manuscript has been judged scientifically suitable for publication and will be formally accepted for publication once it meets all outstanding technical requirements.

Kind regards,

Ana Cristina Mafla

Academic Editor

PLOS ONE

---

## [Editor Report · Acceptance letter]

PONE-D-25-02377R2

PLOS ONE

Dear Dr. Faytong-Haro,

I'm pleased to inform you that your manuscript has been deemed suitable for publication in PLOS ONE. Congratulations! Your manuscript is now being handed over to our production team.

Kind regards,

on behalf of

Dr. Ana Cristina Mafla

Academic Editor

PLOS ONE